# Roles of Small Polyetherimide Moieties on Thermal Stability and Fracture Toughness of Epoxy Blends

**DOI:** 10.3390/polym13193310

**Published:** 2021-09-28

**Authors:** Seul-Yi Lee, Min-Joo Kang, Seong-Hwang Kim, Kyong Yop Rhee, Jong-Hoon Lee, Soo-Jin Park

**Affiliations:** 1Department of Chemistry, Inha University, 100 Inharo, Incheon 22212, Korea; leesy1019@inha.ac.kr (S.-Y.L.); sa-rahlunar@naver.com (M.-J.K.); seonghwangkim@inha.edu (S.-H.K.); boy834@naver.com (J.-H.L.); 2Department of Mechanical Engineering, College of Engineering, Kyung Hee University, 1732 Deogyeong-daero, Yongin 17104, Korea

**Keywords:** epoxy resins, polyetherimide, thermal stability, fracture toughness

## Abstract

Bisphenol A diglycidyl ether (DGEBA) was blended with polyetherimide (PEI) as a thermoplastic toughener for thermal stability and mechanical properties as a function of PEI contents. The thermal stability and mechanical properties were investigated using a thermogravimetric analyzer (TGA) and a universal test machine, respectively. The TGA results indicate that PEI addition enhanced the thermal stability of the epoxy resins in terms of the integral procedural decomposition temperature (IPDT) and pyrolysis activation energy (*E*_t_). The IPDT and *E*_t_ values of the DGEBA/PEI blends containing 2 wt% of PEI increased by 2% and 22%, respectively, compared to those of neat DGEBA. Moreover, the critical stress intensity factor and critical strain energy release rate for the DGEBA/PEI blends containing 2 wt% of PEI increased by 83% and 194%, respectively, compared to those of neat DGEBA. These results demonstrate that PEI plays a key role in enhancing the flexural strength and fracture toughness of epoxy blends. This can be attributed to the newly formed semi-interpenetrating polymer networks (semi-IPNs) composed of the epoxy network and linear PEI.

## 1. Introduction

Epoxy is extensively used as a typical thermosetting resin. Epoxy resin is widely used as adhesives, high-performance coatings, packaging, and lamination materials because it exhibits excellent chemical stability, low shrinkage, good adhesion to various metals, good thermal stability, and good mechanical properties [1,2,3,4,5,6]. However, from the perspective of high-performance engineering applications, pure epoxy resin possesses some limitations, including poor crack resistance and inherent brittleness with the high cross-linking density for curing. To overcome these problems, several methodologies, including the integration of a second phase (e.g., thermoplastics and inorganic nonparties), have been explored to improve the fracture toughness and strength in modified epoxy systems. Among the methodologies, the formation of an interpenetrating polymer network (IPN) via in situ polymerization of the toughener component has gained significant attention for obtaining a synergistic combination of desirable properties of two and/or more polymers [7]. IPN are special blends defined as mixtures of two or more cross-linked polymer networks with intermolecular interlocking, rather than chemical bonds. The network interlocking occurs during the curing process and affects the physical properties of the blends.

Since the term of “IPN” was first introduced by John Millar in 1960, various studies have reported about IPN-based epoxy systems incorporating thermoplastic polymers, such as polysurfone (PSF), polyethersulfone (PES), polycarbonate (PC), polyetheretherketone (PEEK), and polyetherimide (PEI), and have demonstrated the enhancement in mechanical properties (i.e., fracture toughness) of epoxy resins [8,9,10].

Lee et al. [11] reported that the fracture toughness of PES/epoxy blends increased by up to 11% due to the formation of semi-IPN structures. Wang et al. [12] demonstrated the synthesis of a toughening agent based on a hyper-branched polyester with flexible chain blocking. The toughening agent improved the bending strength and impact strength of the blends by 122% and 184%, respectively, which was associated with the formation of IPN between the flexible chain of the toughener and epoxy matrix. Liu et al. [13] prepared semi-IPN blends of epoxy and polyimide without the aid of any solvents and exhibited improved processability, as evidenced by the low viscosity onset temperature and reduced gelatin time at 200 °C. Furthermore, due to the newly formed strong networks (semi-IPN) in the epoxy system, outstanding mechanical properties were displayed: impact strength of 22–63 kJ·m^−2^, flexural strength of 156–172 MPa, and tensile strength of 78–85 MPa.

PEI is an amorphous engineering thermoplastic, and compared to other polyimides, it affords a higher glass transition temperature (*T*_g_) of ≈215 °C, higher strength and rigidity at elevated temperatures, higher moisture resistance, lower shrinkage, etc., thereby showing a wider range of processing capabilities. Recently, PEI has emerged as a promising candidate for enhancing the mechanical properties of epoxy systems for engineering applications due to its good compatibility and miscibility with epoxy resins. PEI has been reported to enhance the interfacial adhesion between the two phases by forming a strong network, i.e., semi-IPN structures, which increases the thermal and mechanical properties of the resultant epoxy system. These improve the stress transfers at the interfaces between PEI and epoxy without significantly decreasing the modulus or *T*_g_ and resistance to solvents and radiation [14,15,16,17].

In this study, DGEBA/PEI blends were prepared with varying PEI content. PEI was added to epoxy resin as a thermoplastic toughener to investigate mechanical behaviors through flexural strength and fracture toughness testing in the prepared blends. The integral procedural decomposition temperature (IPDT) and pyrolysis activation energy (*E*_t_) values of the DGEBA/PEI blends were higher than those of neat epoxy by 2% and 22%, respectively. Furthermore, after the introduction of 2 wt% of PEI into the epoxy, the *K*_IC_ and *G*_IC_ values increased by 83% and 194%, respectively. Our experiment confirms that the thermal stability and mechanical properties of epoxy can be improved by adding PEI as a toughener. Thus, these results imply a great potential of PEI-loaded epoxy blends for the applications in civil and structural engineering. We believe that this work provides a guidance for utilizing thermoplastic polymers, which could be a powerful strategy to extend the tuning range in the future design and manufacturing of epoxy blends.

## 2. Materials and Methods

### 2.1. Materials

Bisphenol A diglycidyl ether (DGEBA; YD 128) as the epoxy matrix was supplied by Kukdo Chemical Co. (Seoul, Korea) its epoxide-equivalent weight was 185–190 g·equiv^−1^. PEI as a thermoplastic toughener was supplied by Sigma-Aldrich Co. (St. Louis, MO, USA), with average molecular weights of Mw = 25,000 and Mn = 10,000, viscosity of 15,000 mPa·s at 50 °C, and density of 1.27 g·cm^−3^ at 25 °C. 4,4′-Diaminodiphenylmethane (DDM) and methylene chloride (MC), as the curing agent and solvent, respectively, were obtained from Sigma-Aldrich Co. The chemical structures of DGEBA, PEI, and DDM are shown in Figure 1.

### 2.2. Preparation of Blends

DGEBA/PEI blends were prepared with a weight ratio of 1–4 wt%. PEI was dissolved in MC by stirring at room temperature for 4 h. DGEBA was mixed with the PEI solution at 100 °C for 24 h, and the mixture was placed in a vacuum oven to evaporate the solvent. This mixture was additionally stirred at 60 °C to completely remove the bubbles and residual solvent. DDM was added into the mixture, and then, the final mixture was injected into a mold. The obtained blends were transferred to a convection oven for curing at 110 °C for 1 h, 140 °C for 2 h, and 170 °C for 1 h. Figure 2 illustrates the preparation method for the DGEBA/PEI blends.

### 2.3. Characterization and Measurements

Fourier transform-infrared (FT-IR, BRUKER VERTEX 80 V, BRUKER, Billerica, MA, USA) spectra of DGEBA/PEI blends were obtained by background subtraction and ATR corrected in the range from 800 to 4000 cm^−1^. The thermal stability of DGEBA/PEI blends was analyzed using a thermogravimetric analyzer (TGA, NETZSCH TG209 F3, ETZSCH, Selb, Germany). The analysis was conducted under nitrogen flow with increasing temperature from 50 to 800 °C at a heating rate of 10 °C·min^−1^. The critical stress intensity factor (*K*_IC_ ), critical strain energy release rate (*G*_IC_), and fracture toughness of the prepared specimen were measured using a universal test machine according to ASTM D5045-95. The sample size of the single edge notch specimen was 5 × 10 × 50 mm^3^, and the cross-head speed was 10 mm·min^−1^. After performing the *K*_IC_ fracture toughness tests, the fractured surfaces were observed using a scanning electron microscope (SEM, HITACHI SU8010) to investigate the DGEBA/PEI blend morphology.

## 3. Results and Discussion

### 3.1. Curing of the DGEBA/PEI Blends

Figure 3a shows the FT-IR spectra of the DGEBA epoxy resin, cured neat DGEBA, and DGEBA/PEI blends containing 2 wt% of PEI. In addition, the FT-IR spectra of the PEI is given in Appendix A. The characteristic absorption peaks of the epoxide and hydroxyl groups appeared at 906 and 3429 cm^−1^, respectively [18,19]. For the DGEBA/PEI blends, new peaks were observed at 1716 cm^−1^, which can be attributed to stretching vibrations of the imide group (typical of imide carbonyl asymmetrical and symmetrical stretch). The peaks of the epoxide group for the cured DGEBA and DGEBA/PEI blends significantly decreased after the curing reaction. Moreover, the peaks of the hydroxyl stretching vibrations increased due to the reaction between the amine group in DDM and the epoxide group to form a hydroxyl group. Partial spectra of the neat DGEBA and the DGEBA/PEI blends in the 3000–3800 cm^−1^ wavenumber range are shown in Figure 3b. The peak of the hydroxyl stretching vibrations for the DGEBA/PEI blends shifted to lower wavenumbers after PEI addition. This suggests that intermolecular interactions, such as hydrogen bonding, occurred between the hydroxyl groups in DGEBA and carbonyl groups in PEI [20,21,22,23,24].

## 3.2. Thermal Stability of the DGEBA/PEI Blends

Thermal stability is a critical factor on polymeric materials for processing and practical application. Figure 4a shows the thermogravimetric curves of the neat DGEBA and DGEBA/PEI blends. As expected, the significant changes of the thermal stability studied in the work were not apparent in the presence of small moieties of PEI in the epoxy blend. However, it is found that the variation of the activation energy of decomposition can be revealed in the work, as shown in Figure 4b, degradation occurred between 360 and 420 °C. The thermal stability factors, including the polymer decomposition temperature (PDT), IPDT, temperatures of maximum rate of degradation (*T*_max_), and *E*_t_, were calculated from the thermogravimetric results [25]. For the delineation of dynamics of thermal decomposition, kinetic parameters of IPDT that correlate with the volatile components were determined from Doyle’s proposition as follows [26]:(1)$$\mathrm{IPDT}=A^{*}\times K^{*}\times \left(T_{f}-T_{i}\right)+T_{i}$$
(2)$$A^{*}=\frac{\left(S_{1}+S_{2}\right)}{\left(S_{1}+S_{2}+S_{3}\right)}$$
(3)$$K^{*}=\frac{\left(S_{1}+S_{2}\right)}{S_{1}}$$
where *A^*^* is the area ratio of the total experimental curve defined by the total TGA thermogram, *K^*^* is the coefficient of *A^*^*, *T*_i_ is the initial experimental temperature, and *T*_f_ is the final experimental temperature. The inherent thermal stability of polymer blends during thermal degradation is understood by IPDT.

For detailed investigation of the polymer blends’ decomposition and activation energy of thermal degradation, the (*E*_t_) were calculated from the TGA curves using the Horowitz–Metzger method as follows [27]:
(4)$$ln\left[ln{\left(1-\alpha \right)}^{-1}\right]=\frac{E_{t}\theta }{RT_{\max}^{2}}$$
(5)$$\alpha =\frac{W_{0}-W_{t}}{W_{0}-W_{\infty }}$$
where *α* is the decomposition fraction, $\theta =T-T_{\max}$, *R* is the universal gas constant, *W*_t_ is the actual mass of the sample, *W*_0_ is the initial mass of the sample, and *W*_∞_ is the final mass of the sample. Figure 4b shows the plots of *ln*[*ln(*1 – *α*)^–1^] vs. *θ*. The *E*_t_ value was calculated from the slope of the straight lines using Equation (4).

Table 1 summarizes the calculated PDT, IPDT, *T*_max_, and *E*_t_ values of the neat DGEBA and DGEBA/PEI blends. The PDT and IPDT values of the blends slightly increased due to PEI addition. The *T*_max_ value of the DGEBA/PEI blends was similar to that of neat DGEBA. The *E*_t_ value of neat DGEBA was 256.3 kJ·mol^−1^, whereas that of the blends was significantly higher (294.1–312.8 kJ·mol^−1^). This confirms that more activation energy was needed for pyrolysis, as the DGEBA/PEI blend was more stable and more difficult to decompose [28,29,30].

## 3.3. Mechanical Properties of the DGEBA/PEI Blends

Flexural strength (σ
) and flexural modulus ($E_{b}$) of the DGEBA/PEI blends with different PEI loading amounts were also calculated, using the following Equations (6) and (7):
(6)$$\sigma =\frac{3PL}{2bd^{2}}$$
(7)$$E_{b}=\frac{L^{3}}{4{\mathrm{bd}}^{3}}\frac{∆P}{∆m}$$
where P
is the applied peak load, b
is the sample width, d
is the specimen thickness, L
is the support span, ∆P
is the variation in force at the linear portion, and ∆m
is the relative deflection variation.

Figure 5 shows the flexural strength and flexural modulus of DGEBA/PEI blends. The σ
and $E_{b}$
values of the blends increased with the PEI content up to 2 wt%. σ
and $E_{b}$
values of the blends containing 2 wt% of PEI were 143.3 MPa and 2.8 GPa, respectively, which were 20% and 17% higher than those of neat DGEBA (119.7 MPa and 2.4 GPa).

The flexural strengths of DGEBA-based blends are compared in Table 2. The optimized DGEBA/PEI sample appeared to have superior flexural strength compared to other DGEBA-based blends. It appears that the PEI plays an important role as a mechanical reinforcement in improving the flexural strength, resulting from enhanced semi-IPN between the PEI and epoxy matrix.

Fracture toughness is a fundamental property of polymeric materials indicating the strain energy and absorbing ability of pre-fracture materials. A higher fracture toughness indicates higher resistance to crack propagation. The fracture toughness of the DGEBA/PEI blends was investigated using the *K*_IC_ and *G*_IC_ values (Figure 5). The *K*_IC_ and *G*_IC_ values can be calculated as follows [37,38,39]:
(8)KIC=PBW12fa/w
(9)$$f\left(a/W\right)=\frac{\left(2+a/W\right)\;\left[0.886\;+4.64a/W\;-\;13.32a/W^{2}\;+\;14.72a/W^{3}\;-\;5.6a/W^{4}\right)]}{{\left(1-a/W\right)}^{3/2}}$$
where *P* (N), *a* (mm), *W*, and *B* are the critical load, crack length, width, and thickness of the specimen, respectively. Moreover, *f* (*a*/*W*) is the geometric element obtained from Equation (7).
(10)$$G_{\mathrm{IC}}=\frac{\left(1-v^{2}\right)\cdot K_{\mathrm{IC}}^{2}}{E}$$
where *υ* is the Poisson’s ratio (0.3), and *E* is the tensile modulus obtained from the fracture testing [40].

Figure 6 shows the fracture toughness of the DGEBA/PEI blends as a function of the PEI content. The load–deflection curves of DGEBA/PEI blends are shown in Figure 6a. It can be seen that neat DGEBA fractured at low deflection before yielding, whereas the DGEBA/PEI blends exhibited ductile behavior with yielding and subsequent plastic deformation. The *K*_IC_ and *G*_IC_ values of the blends increased with the PEI content up to 2 wt%. The *K*_IC_ values of the blends containing the blends containing 2 wt% of PEI were 4.9 MPa·m^1/2^, which were 83% higher than those of neat DGEBA (Figure 6b). Moreover, *G*_IC_ values of the blends containing 2 wt% of PEI were 58.5 kJ·m^−2^, which were 194% higher than those of neat DGEBA (Figure 6c). This can be attributed to the newly formed intermolecular interaction (semi-IPN) between PEI and the epoxy matrix in the DGEBA/PEI blends. The mechanical properties, including the *K*_IC_ and *G*_IC_, of the DGEBA/PEI blends are indicated in Figure 6d. Here, all the DGEBA/PEI blends exhibit excellent linear relationships between the *K*_IC_ and the *G*_IC_. Thus, we confirmed that the PEI particles in the epoxy matrix act as stress concentrators to absorb the external energy and mitigate the crack growth [41,42,43,44,45,46].

Figure 7 shows SEM images of the morphologies of the fractured surfaces for neat DGEBA and DGEBA/PEI blends after the fracture tests. It indicates the crack progression on the fracture surfaces. As shown in Figure 7a,d, neat DGEBA displayed a regular fracture surface, which is the typical characteristic of a brittle fracture. In contrast, the blends containing 2 wt% of PEI (Figure 7b,e) showed rough surfaces, meaning a ductile fracture. However, the blends containing 4 wt% of PEI (Figure 7c,f) were shown in the structure of the separated phases. From the results, it was found that excellent semi-IPN structures were formed with the blends containing 2 wt % of PEI, which is closely related to the thermal stability and mechanical properties [47,48].

## 4. Conclusions

In this study, we observed the thermal stability and fracture toughness of neat DGEBA and DGEBA/PEI blends. The PDT and IPDT values of the blends slightly increased with the PEI content. The *E*_t_ value of the blends significantly increased by 22% from 256.3 to 312.8 kJ·mol^−1^ after the addition of 2 wt% of PEI. The *K*_IC_ and *G*_IC_ values of the blends containing 2 wt% of PEI were 83% and 194%, respectively, higher than those of neat DGEBA, which is associated with the increase in the activation energy of decomposition in thermal stability studies. The SEM images indicated that the surfaces of the DGEBA/PEI blends were rougher than those of neat DGEBA, accounting for their excellent fracture toughness. In summary, the addition of PEI to DGEBA significantly improved the fracture toughness due to the newly formed semi-IPN structures, which resulted from the combination of two noncompatible polymers. Thus, PEI plays an important role in enhancing the interfacial adhesion of epoxy. Therefore, the applications of the obtained DGEBA/PEI blends should be extended, such as heat resistant adhesive, surface coating, high performance blends, and electrical and electronic encapsulation.

## Figures and Tables

**Figure 1 polymers-13-03310-f001:**
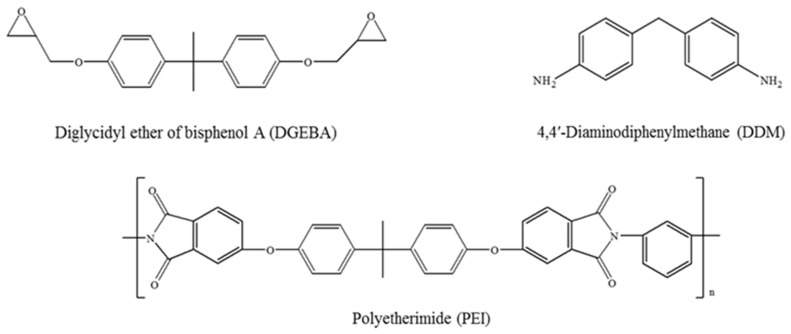
Chemical structures of DGEBA, PEI, and DDM.

**Figure 2 polymers-13-03310-f002:**
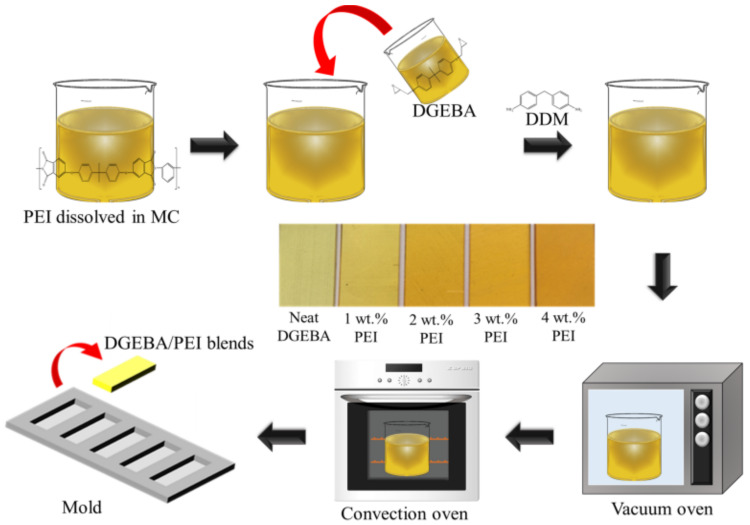
Schematic diagram for the preparation of DGEBA/PEI blends.

**Figure 3 polymers-13-03310-f003:**
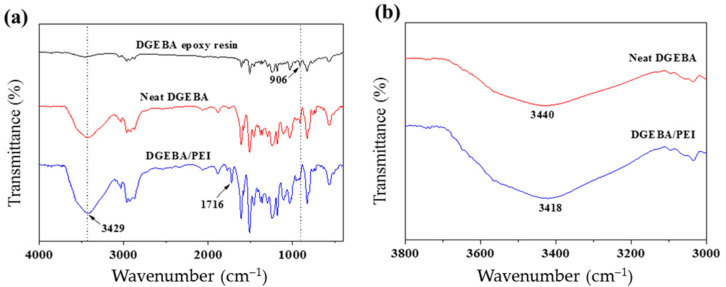
Curing of the DGEBA/PEI blends: (**a**) FT-IR spectra of DGEBA, cured neat DGEBA, and DGEBA/PEI blends containing 2 wt% of PEI and (**b**) Partial FT-IR spectra of DGEBA/PEI blends in the range of 3000–3800 cm^−1^.

**Figure 4 polymers-13-03310-f004:**
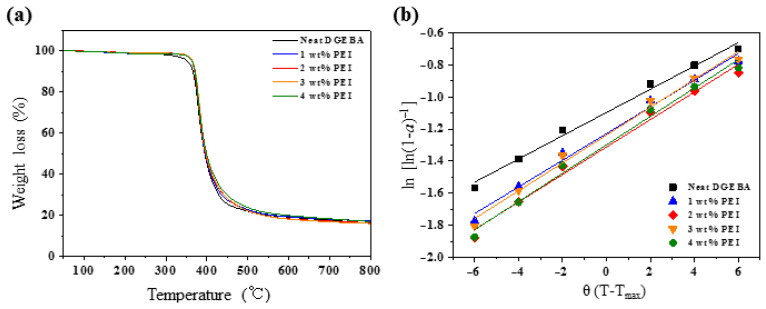
Thermal stability of the DGEBA/PEI blends: (**a**) TGA curves and (**b**) plots of *ln*[*ln*(1 *− α*)^−1^] vs. *θ* using the Horowitz–Metzger method [25].

**Figure 5 polymers-13-03310-f005:**
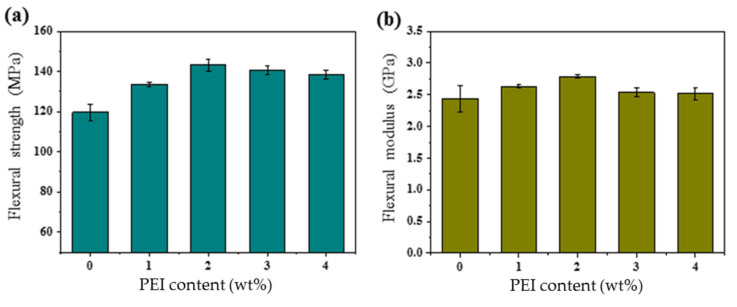
Flexural strengths of the DGEBA/PEI blends: (**a**) flexural strength and (**b**) flexural modulus.

**Figure 6 polymers-13-03310-f006:**
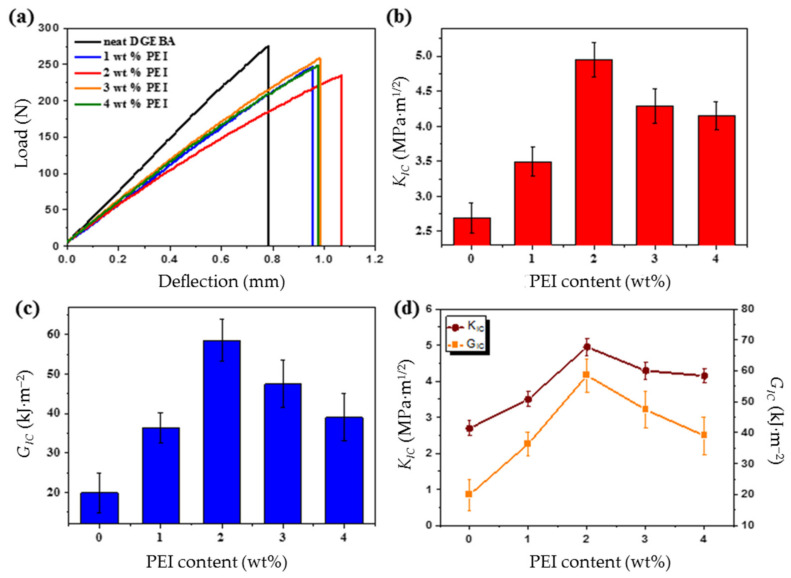
Fracture toughness behaviors of the DGEBA/PEI blends: (**a**) load–deflection curves, (**b**) *K*_IC_, (**c**) *G*_IC_, and (**d**) correlations between the *K*_IC_ and *G*_IC_.

**Figure 7 polymers-13-03310-f007:**
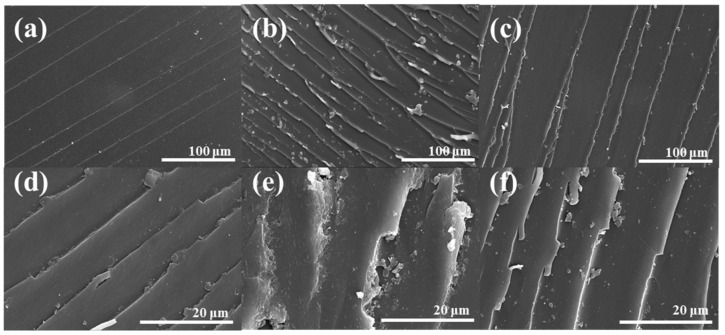
Fracture surface of the DGEBA/PEI blends: (**a**,**d**) neat DGEBA, (**b**,**e**) 2wt % of PEI, and (**c**,**f**) 4 wt% of PEI.

**Table 1 polymers-13-03310-t001:** Thermal stability parameters of the DGEBA/PEI blends.

Specimen	^1^ PDT (°C)	^2^ *A^*^K^*^*	^3^ IPDT (°C)	^4^*T*_max_ (°C)	^5^*E*_t_ (kJ·mol^−1^)
neat DGEBA	364.1	0.81	399.5	379.9	256.3
1 wt% of PEI	366.3	0.82	401.7	379.9	294.0
2 wt% of PEI	366.8	0.86	404.4	379.9	312.8
3 wt% of PEI	366.7	0.84	402.6	379.9	306.9
4 wt% of PEI	366.6	0.81	401.8	379.9	303.3

^1^ PDT: Polymer decomposition temperature; ^2^
*A^*^K ^*^*: Thermal stability index; ^3^ IPDT: Integral procedural decomposition temperature; ^4^
*T*_max_: Temperatures of maximum rate of degradation; ^5^
*E*_t_: Activation energy of decomposition.

**Table 2 polymers-13-03310-t002:** Comparisons of DGEBA-based blends in flexural strength.

Specimen	Content (wt%)	Flexural Strength (MPa)	References
DGEBA/HER-HT	12	84.0	[31]
DGEBA/MHHPA	70	131.6	[32]
DGEBA/D230	30	121.0	[33]
DGEBA/castor oil	20	100.1	[34]
DGEBA/soybean oil	10	127.0	[35]
DGEBA/cardanol epoxy	20	80.8	[36]
DGEBA/PEI	2	143.3	This work

## Data Availability

Not applicable.

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
