# Peer review of "Roles of Small Polyetherimide Moieties on Thermal Stability and Fracture Toughness of Epoxy Blends"

_polymers, 2021, doi:10.3390/polym13193310_

Round 1

Reviewer 1 Report

Major comments authors are listed below:

  1. The introduction should extend with more details.
  2. In the end of introduction, author should report the novelty and the applications of this work.
  3. In Section: 2.3: FTIR analyser in Lines 97 and 98, needs further details.
  4. The interpretation of results is insufficient and authors should be discussed with more information for comparison the results given prior related works.
  5. To improve the quality of this work, mechanical tests should perform.

Author Response

[Reviewer #1]

  1. The introduction should extend with more details.

è Thanks for your valuable comment for enhancing the quality of our paper. We newly added the importance of PEI in Introduction according to the Reviewer’s comment.

< Revised or added paragraphs >

Introduction: Page 1, Line 41-43;

IPN are special blends defined as mixtures of two or more cross-linked polymer networks with intermolecular interlocking, rather than chemical bonds. The network interlocking occurs during curing and affects the physical properties of the blends.

Introduction: Page 2, Line 69-71;

These improve the stress transfers at the interfaces between PEI and epoxy without significantly decreasing the modulus or Tg and resistance to solvents and radiation [14-17].

Introduction: Page 2, Line 72-74;

PEI was added to epoxy resin as a thermoplastic toughener to investigate thermal stability and mechanical behavior through flexural strength and fracture toughness testing in the prepared blends.

  1. In the end of introduction, author should report the novelty and the applications of this work.

è We added some sentences on the novelty and the potential applications of our work in the revised manuscript. Thank you for your suggestion.

< Revised or added paragraph >

Introduction: Page 2, Line 72-83;

In this study, DGEBA/PEI blends were prepared with varying PEI content. PEI was added to epoxy resin as a thermoplastic toughener to investigate mechanical behaviors through flexural strength and fracture toughness testing in the prepared blends. The integral procedural decomposition temperature (IPDT) and pyrolysis activation energy (Et) values of the DGEBA/PEI blends were higher than those of neat epoxy by 2% and 22%, respectively. Furthermore, after the introduction of 2 wt.% of PEI into the epoxy, the KIC and GIC values increased by 83% and 194%, respectively. Our experiment confirms that the thermal stability and mechanical properties of epoxy can be improved by adding PEI as a toughener. Thus, these results imply a great potential of PEI-loaded epoxy blends for the applications in civil & structural engineering. We believe that this work provides a guidance for utilizing thermoplastic polymers, which could be a powerful strategy to extend the tuning range in the future design and manufacturing of epoxy blends.

  1. Section: 2.3: FTIR analyser in Lines 97 and 98, needs further details.

è Thanks for your professional advice. We added the FT-IR details in Section 2.3. Characterization and Measurements.

< Revised or added paragraph >

Materials and Methods: Page 3, Line 107-109;

Fourier transform-infrared (FT-IR, BRUKER VERTEX 80 V) spectra of DGEBA/PEI blends were obtained by background subtraction and ATR corrected in the range from 800 to 4000 cm−1.

  1. The interpretation of results is insufficient and authors should be discussed with more information for comparison the results given prior related works.

è We newly added “Table 2” to compare out result to other reports for DGEBA-based blends in the flexural strength in the revised manuscript.

< Added paragraph and Table >

Results and Discussion: Page 6-7, Line 198-203;

Flexural strengths of DGEBA-based blends are compared in Table 2. The optimized DGEBA/PEI sample appeared superior flexural strength compared to other DGEBA-based blends. It appears that the PEI plays an important role as a mechanical reinforcement in improving the flexural strength, resulting from enhanced semi-IPN between the PEI and epoxy matrix.

Table 2. Comparisons of DGEBA-based blends in flexural strength.

Specimen

Content (wt.%)

Flexural strength (MPa)

References

DGEBA/HER-HT

12

84.0

[31]

DGEBA/MHHPA

70

131.6

[32]

DGEBA/D230

30

121.0

[33]

DGEBA/castor oil

20

100.1

[34]

DGEBA/soybean oil

10

127.0

[35]

DGEBA/cardanol epoxy

20

80.8

[36]

DGEBA/PEI

2

143.3

This work

  1. To improve the quality of this work, mechanical tests should perform.

è We added flexural strength and load-deflection curves in the revised manuscript according to the reviewer’s suggestion.

<Revised or added paragraph and figure>

Results and Discussion: Page 6, Line 183-197;

Flexural strength () and flexural modulus () of the DGEBA/PEI blends with different PEI loading amounts were also calculated, using the following equations (6) and (7):

                                                                                                    (6)

                                                                                                     (7)

where  is the applied peak load,  is the sample width,  is the specimen thickness,  is the support span,  is the variation in force at the linear portion, and  is the relative deflection variation.

Figure 5 shows the flexural strength and flexural modulus of DGEBA/PEI blends. The  and  values of the blends increased with the PEI content up to 2 wt.%.  and  values of the blends containing 2 wt.% of PEI were 143.3 MPa and 2.8 GPa, respectively, which were 20% and 17% higher than those of neat DGEBA (119.7 MPa and 2.4 GPa).

Figure 5. Flexural strength of the DGEBA/PEI blends: (a) flexural strength and (b) flexural modulus.

Results and Discussion: Page 7, Line 219-236;

Figure 6 shows the fracture toughness of the DGEBA/PEI blends as a function of the PEI content. The load-deflection curves of DGEBA/PEI blends are shown in Figure 6(a). It can be seen that neat DGEBA fractured at low deflection before yielding, whereas the DGEBA/PEI blends exhibited ductile behavior with yielding and subsequent plastic deformation. The KIC and GIC values of the blends increased with the PEI content up to 2 wt.%. The KIC values of the blends containing the blends containing 2 wt.% of PEI were 4.9 MPa m1/2, which were 83% higher than those of neat DGEBA (Figure 6(b)). Moreover, GIC values of the blends containing 2 wt.% of PEI were 58.5 kJ m−2, which were 194% higher than those of neat DGEBA (Figure 6(c)). This can be attributed to the newly formed intermolecular interaction (semi-IPN) between PEI and the epoxy matrix in the DGEBA/PEI blends. The mechanical properties, including the KIC and GIC, of the DGEBA/PEI blends are indicated in Figure 6(d). Here, all the DGEBA/PEI blends exhibit excellent linear relationships between the KIC and the GIC. Thus, we confirmed that the PEI particles in the epoxy matrix act as stress concentrators to absorb the external energy and mitigate the crack growth [33-38].

Figure 6. Fracture toughness of the DGEBA/PEI blends: (a) load-deflection curves, (b) KIC, (c) GIC and (d) correlation between KIC and GIC.

.

Reviewer 2 Report

Comments are in the attachment.

Author Response

16] Buketov, A.; Maruschak, P.; Sapronov, O.; Zinchenko, D.; Yatsyuk, V.; Panin, S., Enhancing Performance Characteristics of Equipment of Sea and River Transport by Using Epoxy Composites. Transport-Vilnius 2016, 31, (3), 333-342.

[17] Buketov, A.; Smetankin, S.; Maruschak, P.; Yurenin, K.; Sapronov, O.; Matvyeyev, V.; Menou, A., New Black-Filled Epoxy Coatings for Repairing Surface of Equipment of Marine Ships. Transport-Vilnius 2020, 35, (6), 679-690.

  1. The article does not describe specimens for testing on KIC. There is no description of their geometry and test procedure (loading rate etc.). It is not clear how the GIC was measured. It is desirable to indicate the accuracy of the test parameters.

è We added the test standard and specimen sizes of the samples used in this paper. Please see page 4/Line 112-117.

<Revised or added paragraph and figure>

Materials and Methods: Page 4, Line 112-117;

The critical stress intensity factor (KIC), critical strain energy release rate (GIC), and fracture toughness of the prepared specimen were measured using a universal test machine according to ASTM D5045-95. The sample size of the single edge notch specimen was 5 ´ 10 ´ 50 mm3, and the cross-head speed was 10 mm min−1. After performing the KIC fracture toughness tests, the fractured surfaces were observed using a scanning electron microscope (SEM, HITACHI SU8010) to investigate the DGEBA/PEI blend morphology.

  1. It would be useful to show the loading curves of the specimens tested at KIC.

è We really appreciate your time. We newly added load-deflection curves in the revised manuscript according to the reviewer’s suggestion.

<Revised or added paragraph and figure>

Results and Discussion: Page 7-8, Line 218-234;

Figure 6 shows the fracture toughness of the DGEBA/PEI blends as a function of the PEI content. The load-deflection curves of DGEBA/PEI blends are shown in Figure 6(a). It can be seen that neat DGEBA fractured at low deflection before yielding, whereas the DGEBA/PEI blends exhibited ductile behavior with yielding and subsequent plastic deformation.

Figure 6. Fracture toughness of the DGEBA/PEI blends: (a) load-deflection curves, (b) KIC, (c) GIC and (d) correlation between KIC and GIC.

  1. At SEM analysis (Fig. 6), micromechanisms are very poorly analyzed and systematized. I recommend deepening this part of the description.

è We re-organized the SEM images for the fracture surfaces of the DGEBA/PEI blends and revised the paragraph for the new Figure 7.

< Revised or added paragraph and figure>

Results and Discussion: Page 8, Line 235-246;

Figure 7 shows SEM images of the morphologies of the fractured surfaces for neat DGEBA and DGEBA/PEI blends after the fracture tests. It indicates the crack progression on the fracture surfaces. As shown in Figure 7(a,d), neat DGEBA displayed a regular fracture surface, which is the typical characteristic of a brittle fracture. In contrast, the blends containing 2 wt.% of PEI (Figure 7(b,e)) showed rough surfaces, meaning a ductile fracture. However, the blends containing 4 wt.% of PEI (Figure 7(c,f)) showed in the structure of the separated phases. From the results, it was found that an excellent semi-IPN structures were formed at the blends containing 2 wt.% of PEI, which is closely related to the thermal stability and mechanical properties [47, 48].

Figure 7. Fracture surface of the DGEBA/PEI blends: (a,d) neat DGEBA, (b,e) 2 wt.% of PEI, and (c,f) 4 wt.% of PEI.

Reviewer 3 Report

Presented work is interesting, but should be completed in some points:

  • Dynamic Mechanical Thermal Analysis (DMTA) should be performed. Storage modulus, loss modulus and damping factor vs. temperature should be determined. Cross-link density can be calculated on the basis of DMTA results. Moreoved, DMTA permit to determine the glass transition temperature. Mentioned parameters will be hepfull to characterize the viscoelastic properties of obtained materials.
  • Figure 3: Authors should add the FTIR spectrum of PEI. All spectra should be analysed in details (including intensiy changes of epoxy groups). Presence of imide moieties should be also confirmed
  • Some characteristic of PEI should be presented (average molecular weight, functionality, viscosity). Is PEI terminated by amine groups?
  • Table 1 should be completed by Td5% and Td10% values.
  • Figure 6: SEM micrographs should be enlarged
  • Why PEI improve fracture thoughness? Authors should also determine the tensile properties (including tensile modulus)

In my opinion presented work should be reconsidered after major revisions (the main important are DMTA results).

Author Response

[Reviewer #3]

  1. Dynamic Mechanical Thermal Analysis (DMTA) should be performed. Storage modulus, loss modulus and damping factor vs. temperature should be determined. Cross-link density can be calculated on the basis of DMTA results. Moreover, DMTA permit to determine the glass transition temperature. Mentioned parameters will be helpfull to characterize the viscoelastic properties of obtained materials.

è Thanks for your kind comment. This study aimed to study the roles of PEI moieties on thermal stability and fracture toughness in the epoxy blend systems. We carefully evaluated thermal stability of samples by polymer decomposition temperature (PDT), integral procedural decomposition temperature (IPDT), maximum rate of degradation (Tmax), and activation energy of thermal degradation (Et). Obvious enhancement of thermal stability was shown by increase of PDT and IPDT with the introduction of PEI moieties. Also, the effects of PEI moieties were studied by the thermal kinetics of the blends; the higher Et was shown by addition of PEI. (please see Figure 4 and Table 1)

Fracture toughness can be indicative of the fundamental characteristics of polymeric materials, e.g., the strain energy and absorbing ability of pre-fracture materials. We observed the improvement of fracture toughness in the PEI/epoxy blends (Figure 5 and Table 2). Therefore, we believe these results have provided sufficient explanation for roles of PEI with various analysis methods. Thanks for your precious time and valuable comments again.

  1. Figure 3: Authors should add the FTIR spectrum of PEI. All spectra should be analysed in details (including intensity changes of epoxy groups). Presence of imide moieties should be also confirmed

è We really appreciate your time. We added FT-IR of the PEI in Supporting Information. In addition, we revised FT-IR spectrum (imide moieties) in the manuscript according to the reviewer’s suggestion.

<Revised or added paragraph and figure>

Results and Discussion: Page 4, Line 120-127, 136-138;

Figure 3(a) shows the FT-IR spectra of the DGEBA epoxy resin, cured neat DGEBA, and DGEBA/PEI blends containing 2 wt.% of PEI. In addition, the FT-IR spectra of the PEI is given in Figure S1. The characteristic absorption peaks of the epoxide and hydroxyl groups appeared at 906 and 3429 cm1, respectively [18, 19]. For the DGEBA/PEI blends, new peaks were observed at 1716 cm1, which can be attributed to stretching vibrations of the imide group (typical of imide carbonyl asymmetrical and symmetrical stretch). The peaks of the epoxide group for the cured DGEBA and DGEBA/PEI blends significantly decreased after the curing reaction.

Figure 3. Curing of the DGEBA/PEI blends: (a) FT-IR spectra of DGEBA, cured neat DGEBA, and DGEBA/PEI blends containing 2 wt.% of PEI; (b) Partial IR spectra of DGEBA/PEI blends in the range of 3000-3800 cm-1.

Supporting Information

Figure S1. FT-IR spectrum of PEI.

  1. Some characteristic of PEI should be presented (average molecular weight, functionality, viscosity). Is PEI terminated by amine groups?

è We added the details of PEI, including average molecular weight, viscosity, and density of reagent in Section 2.1. And the used PEI in this work is terminated by amine groups. Thanks for your precious time and valuable comments again.

<Revised or added paragraph and figure>

Materials and Methods: Page 2, Line 87-90;

PEI as a thermoplastic toughener was supplied by Sigma-Aldrich Co., with average molecular weights of Mw 25,000 and Mn 10,000, viscosity of 15,000 mPa.s at 50 °C, and density of 1.27 g cm−3 at 25°C.

  1. Table 1 should be completed by Td5%and Td10%values.

è When the PEI content reached to 3.0 wt%, the thermal stability and mechanical properties of the DGEBA/PEI blends decreased dramatically. This is because of the excessive bond itself rather than within an epoxy matrix, which is attributable to the intermolecular interaction (semi-IPN) [1-3]. It should be highlighted that, in a reasonable capacity range, a higher value of thermal stability and mechanical properties can be obtained by increasing the PEI contents.

References

[1] Chen, H. L.; You, J. W.; Porter, R. S., Intermolecular interaction and conformation in poly (ether ether ketone)/poly (ether imide) blends—An infrared spectroscopic investigation. Journal of Polymer Research 1996, 3, 151-158.

[2] Zhen, X.; Li, W.; Wu, J.; Jin, X.; Wu, J.; Chen, K.; Gan, W., Effect of tertiary polysiloxane on the phase separation and properties of epoxy/PEI blend. Journal of Applied Polymer Science 2021, 138, 49672.

[3] Gan, W.; Xiong, W.; Yu, Y.; Li, S., Effects of the molecular weight of poly (ether imide) on the viscoelastic phase separation of poly (ether imide)/epoxy blends. Journal of applied polymer science 2009, 114, 3158-3167.

  1. Figure 6: SEM micrographs should be enlarged.

è We added enlarged fracture surfaces for the DGEBA/PEI blend to make it more clear in Figure 7. ​Thanks for your precious time and valuable comments again.

< Revised or added paragraph and figure >

Results and Discussion: Page 8, Line 235-246;

Figure 7 shows SEM images of the morphologies of the fractured surfaces for neat DGEBA and DGEBA/PEI blends after the fracture tests. It indicates the crack progression on the fracture surfaces. As shown in Figure 7(a,d), neat DGEBA displayed a regular fracture surface, which is the typical characteristic of a brittle fracture. In contrast, the blends containing 2 wt.% of PEI (Figure 7(b,e)) showed rough surfaces, meaning a ductile fracture. However, the blends containing 4 wt.% of PEI (Figure 7(c,f)) showed in the structure of the separated phases. From the results, it was found that an excellent semi-IPN structures were formed at the blends containing 2 wt.% of PEI, which is closely related to the thermal stability and mechanical properties [47, 48].

Figure 7. Fracture surface of the DGEBA/PEI blends: (a,d) neat DGEBA, (b,e) 2 wt.% of PEI, and (c,f) 4 wt.% of PEI.

  • Why PEI improve fracture toughness? Authors should also determine the tensile properties (including tensile modulus)

è Thanks for your professional advice. A major factor enhancing fracture toughness can be attributed to the newly formed intermolecular interaction (semi-IPN) between PEI and the epoxy matrix in the DGEBA/PEI blend. To improve the quality of this work, we added flexural strength and load-deflection curves in the revised manuscript.

< Revised or added paragraph and figure >

Results and Discussion: Page 6-7, Line 182-203;

Flexural strength () and flexural modulus () of the DGEBA/PEI blends with different PEI loading amounts were also calculated, using the following equations (6) and (7):

                                                                                                    (6)

                                                                                                    (7)

where  is the applied peak load,  is the sample width,  is the specimen thickness,  is the support span,  is the variation in force at the linear portion, and  is the relative deflection variation.

Figure 5 shows the flexural strength and flexural modulus of DGEBA/PEI blends. The  and  values of the blends increased with the PEI content up to 2 wt.%.  and  values of the blends containing 2 wt.% of PEI were 143.3 MPa and 2.8 GPa, respectively, which were 20% and 17% higher than those of neat DGEBA (119.7 MPa and 2.4 GPa).

Figure 5. Flexural strength of the DGEBA/PEI blends: (a) flexural strength and (b) flexural modulus.

Flexural strengths of DGEBA-based blends are compared in Table 2. The optimized DGEBA/PEI sample appeared superior flexural strength compared to other DGEBA-based blends. It appears that the PEI plays an important role as a mechanical reinforcement in improving the flexural strength, resulting from enhanced semi-IPN between the PEI and epoxy matrix.

Table 2. Comparisons of DGEBA-based blends in flexural strength.

Specimen

Content (wt.%)

Flexural strength (MPa)

References

DGEBA/HER-HT

12

84.0

[31]

DGEBA/MHHPA

70

131.6

[32]

DGEBA/D230

30

121.0

[33]

DGEBA/castor oil

20

100.1

[34]

DGEBA/soybean oil

10

127.0

[35]

DGEBA/cardanol epoxy

20

80.8

[36]

DGEBA/PEI

2

143.3

This work

Results and Discussion: Page 7-8, Line 217-234;

Figure 6 shows the fracture toughness of the DGEBA/PEI blends as a function of the PEI content. The load-deflection curves of DGEBA/PEI blends are shown in Figure 6(a). It can be seen that neat DGEBA fractured at low deflection before yielding, whereas the DGEBA/PEI blends exhibited ductile behavior with yielding and subsequent plastic deformation. The KIC and GIC values of the blends increased with the PEI content up to 2 wt.%. The KIC values of the blends containing the blends containing 2 wt.% of PEI were 4.9 MPa m1/2, which were 83% higher than those of neat DGEBA (Figure 6(b)). Moreover, GIC values of the blends containing 2 wt.% of PEI were 58.5 kJ m−2, which were 194% higher than those of neat DGEBA (Figure 6(c)). This can be attributed to the newly formed intermolecular interaction (semi-IPN) between PEI and the epoxy matrix in the DGEBA/PEI blends. The mechanical properties, including the KIC and GIC, of the DGEBA/PEI blends are indicated in Figure 6(d). Here, all the DGEBA/PEI blends exhibit excellent linear relationships between the KIC and the GIC. Thus, we confirmed that the PEI particles in the epoxy matrix act as stress concentrators to absorb the external energy and mitigate the crack growth [41-46].

Figure 6. Fracture toughness of the DGEBA/PEI blends: (a) load-deflection curves, (b) KIC, (c) GIC, and (d) correlations between KIC and GIC.

  1. In my opinion presented work should be reconsidered after major revisions (the main important are DMTA results).

è Thanks for your kind comment. This study aimed to study the roles of PEI moieties on thermal stability and fracture toughness in the epoxy blend systems. We carefully evaluated thermal stability of samples by polymer decomposition temperature (PDT), integral procedural decomposition temperature (IPDT), maximum rate of degradation (Tmax), and activation energy of thermal degradation (Et). Obvious enhancement of thermal stability was shown by increase of PDT and IPDT with the introduction of PEI moieties. Also, the effects of PEI moieties were studied by the thermal kinetics of the blends; the higher Et was shown by addition of PEI. (please see Figure 4 and Table 1)

Fracture toughness can be indicative of the fundamental characteristics of polymeric materials, e.g., the strain energy and absorbing ability of pre-fracture materials. We observed the improvement of fracture toughness in the PEI/epoxy blends (Figure 5 and Table 2). Therefore, we believe these results have provided sufficient explanation for roles of PEI with various analysis methods. Thanks for your precious time and valuable comments again.

Round 2

Reviewer 1 Report

Comments to authors listed below:

  1. Figure 6 (a): Load (N) vs Deflection (mm) curve should include with mechanical characterizations section (Figure 5). Also, they need to be interpreted in detail and the numerical values need to compare with previous literature.  
  2. The significant numerical values from mechanical test (Flexural strength and modulus) should include in the conclusion section.

Reviewer 2 Report

Accept.

Reviewer 3 Report

Work was improved in comparison to previous submission, so in my opinion it can be accepted in present form. Thank you very much for the provided corrections and completions.